# Trophodynamics of the Antarctic toothfish (*Dissostichus mawsoni*) in the Antarctic Peninsula: Ontogenetic changes in diet composition and prey fatty acid profiles

Karina Pérez-Pezoa[1], César A. Cárdenas[2,3], Marcelo González-Aravena[2], Pablo Gallardo[4], Alí Rivero[4], Vicente Arriagada[5], Kostiantyn Demianenko[6], Pavlo Zabroda[6], Francisco Santa Cruz[2]*

1 Departamento de Ecología, Pontificia Universidad Católica de Chile, Santiago, Chile, 2 Departamento Científico, Instituto Antártico Chileno, Plaza Muñoz Gamero, Punta Arenas, Chile, 3 Millennium Institute Biodiversity of Antarctic and Subantarctic Ecosystems (BASE), Santiago, Chile, 4 Departamento de Ciencias Agropecuarias y Acuícolas, Universidad de Magallanes, Punta Arenas, Chile, 5 Departamento de Microbiología, Universidad de Concepción, Concepción, Chile, 6 Institute of Fisheries and Marine Ecology (IFME), Berdyansk, Ukraine

* fsantacruz@inach.cl

**Data Availability Statement:** All relevant data are within the paper and its Supporting Information files.

## Abstract

The Antarctic toothfish (*Dissostichus mawsoni*) is the largest notothenioid species in the Southern Ocean, playing a keystone role in the trophic web as a food source for marine mammals and a top predator in deep-sea ecosystems. Most ecological knowledge on this species relies on samples from areas where direct fishing is allowed, whereas in areas closed to fishing, such as the Antarctic Peninsula (AP), there are still key ecological gaps to ensure effective conservation, especially regarding our understanding of its trophic relationships within the ecosystem. Here, we present the first comprehensive study of the feeding behavior of Antarctic toothfish caught in the northern tip of the AP, during two consecutive fishing seasons (2019/20 and 2020/21). Stomach content was analyzed according to size-classes, sex and season. Macroscopic morphological analysis was used to identify prey, whereas DNA analysis was used in highly digested prey items. Fatty acid analysis was conducted to determine the prey's nutritional composition. The diet mainly consisted of Macrouridae, Cephalopoda, Anotopteridae, and Channichthyidae. Other prey items found were crustaceans and penguin remains; however, these were rare in terms of their presence in stomach samples. Sex had no effect on diet, whereas size-class and fishing season influenced prey composition. From 27 fatty acid profiles identified, we observed two different prey groups of fishes (integrated by families Anotopteridae, Macrouridae and Channichthyidae) and cephalopods. Our results revealed a narrow range of prey items typical of a generalist predator, which probably consumes the most abundant prey. Understanding the diet and trophic relationships of Antarctic toothfish is critical for a better comprehension of its role in the benthic-demersal ecosystem of the AP, key for ecosystemic fisheries management, and relevant for understanding and predicting the effect of climate change on this species.

**Funding:** Funding for this study was provided by the Marine Protected Areas Program (Number 24 03 052) of the Instituto Antártico Chileno. CAC is also funded by ANID-Millennium Science Initiative Program—ICN2021_002

**Competing interests:** The authors have declared that no competing interests exist

## Introduction

The Southern Ocean is characterized by extreme physical conditions that have shaped a unique, endemic, and highly adapted fauna [1, 2]. An example is its fish fauna, characterized by relatively low species richness and diversity, dominated by the suborder Notothenioidei, a group highly adapted to cold waters [1, 3]. Among this group, *Dissostichus mawsoni* commonly known as Antarctic toothfish (hereafter TOA) is a species with a circumpolar distribution south of 60˚S latitude, inhabiting cold waters (with temperaturas below 0˚C) at depths up to 3000 meters along the shelf and continental slope [4–6]. Individuals can exceed 2 m in length and 100 kg in weight [7, 8], lasting over 30 years, with a first sexual maturation between 12 and 16 years of age [5, 9]. The TOA is by far the largest fish species, playing a key ecological role in the trophic web, both as a food source for marine mammals such as cetaceans [10] and seals [11], and as a top predator in deep-sea ecosystems, structuring the size and population dynamics of prey species through predation [8, 12].

TOA is a valuable fishing resource targeted by an international bottom-set longline fleet that is managed by the Commission for the Conservation of Antarctic Marine Living Resources (CCAMLR), whose main objective is to achieve a balance between the rational use and conservation of fishing stocks. CCAMLR also promotes ecosystem-based fisheries management, through conservation measures based on exploitation levels that ensure recruitment stability and ecological relationships functioning to avoid irreversible changes in the marine ecosystems [13]. At present, TOA is targeted throughout almost their entire distribution range, with a total annual catch of around 4000 tons occurring mostly in the Ross Sea, East Antarctica and the Weddell Sea [14]. The only exception is the Antarctic Peninsula (AP, FAO Statistical Subarea 48.1), where direct fishing for notothenioid is prohibited, due to population collapses after overexploitation in the 1970s and 1980s and still no evidence of population recovery [15]. Nowadays, CCAMLR only allows exploratory fisheries with minimum catch limits aimed to obtain information on the biological and ecological interspecific relationships for ecosystem-based purposes [8, 5, 16, 17].

The prohibition of regular fishing activities has determined knowledge gaps regarding the population and community dynamics of the TOA in Subarea 48.1; thus the CCAMLR Scientific Committee has asked for new studies to reduce uncertainty [18]. Information on trophic dynamics is scarce, and diet analysis based on prey composition is necessary to understand population dynamics and its interaction with the surrounding community [17]. Macroscopic morphological analysis of stomach contents is a widely used technique for prey identification. Despite limitations associated with quick digestion of some fragile prey items, that could be undetected macroscopically, and therefore nutritionally underestimated [19], it can provide useful information on intra- and interspecific predator-prey relationships. Across the Southern Ocean, where fisheries regularly occurs, several dietary studies have been conducted using macroscopic and molecular prey identification [8, 20] with recent studies using a combination of both, providing new information that have improved the understanding of TOA trophic ecology [21]. This type of approach complemented with other methods such as fatty acid analyses (lipids as biomarkers) has proven to be a powerful tools to infer trophic interactions and obtain longer-term information on energy flow through the ecosystem [21–24]. Fatty acids (Fas) are a major source of nutrients and energy in aquatic food webs [25]. FAs can be expressed in relative terms (proportional abundance of the total fatty acid content), where higher proportions of some essential ones, such as eicosapentaenoic (EPA) and

docosahexaenoic (DHA), or the proportion between the polyunsaturated ones (n-3/n-6), can be directly correlated with the nutritive value of a certain prey [26].

Feeding patterns described from TOA stomachs using both macroscopical prey composition and fatty acid profiles [22, 23] have indicated that this species is an opportunistic predator, whose diet depends on the availability of prey in a given habitat [5, 27]. Based on this, current scientific consensus supports the idea of TOA as a generalist predator [18], whose diet composition varies ontogenetically, due to changes in vertical distribution throughout its life history [8, 28]. In addition, considering that its diet can vary regionally, is necessary to evaluate these assumptions in less-explored areas such as the Antarctic Peninsula.

Understanding the feeding ecology of TOA is crucial for assessing its ecological role, which is key for ecosystem-based fishery management to avoid indirect adverse effects. A research program conducted by Ukraine in Subarea 48.1 allowed us to analyze stomachs collected over two consecutive summer seasons (2019/20 and 2020/21). Thus we performed the first comprehensive study of feeding ecology through stomach content and fatty acid analyses. In this study, we identified the prey composition of TOA in the AP using macroscopic guides and DNA for digested prey, tested the effect of size-class length, fishing season, and sex on diet variability, and characterized the nutritional prey contribution from fatty acid profiles. We hypothesized that diet composition changes with size-class length, resulting in an ontogenetic changes in feeding habits.

## Materials and methods

### Study area and sample collection

The stomach contents of 159 TOA individuals were analyzed from catches made by the Ukrainian commercial vessel Calipso in Subarea 48.1 (northern tip of the Antarctic Peninsula, Fig 1). Fishing operations were carried out in February 2020 (n = 89 stomachs) and February 2021 (n = 70 stomachs) using spanish bottom longlines at depths between 924–1560 meters in 2019/20 and between 1075–1371 meters in 2020/21 (S1 Table). The squid *Dosidicus gigas* (cut into rectangular pieces) was used as bait, and therefore were not considered in the subsequent stomach content analysis. Each TOA individual was weighted (in grams, g), sexed and sized (total length in cm) onboard. Extracted stomachs were stored frozen at -20°C and sent for analysis to the Bioresources Laboratory of the Chilean Antarctic Institute (INACH) in Punta Arenas, Chile.

### Stomach content analysis

Whole stomachs were thawed, and content was weighed (mST) to the nearest 0.01 g using an electronic scale. Examination of prey included records of digestion status, taxonomic identification, wet weight (g), length (cm), and FA composition.

Digestion status was noted using a five-point scale: not digested (entire prey), slightly digested (low degree of digestion), moderately digested (intermediate degree of digestion), advanced digested (high degree of digestion but identifiable fleshy remains), and heavily digested (accumulated unidentifiable prey remains). Prey up to moderately digested status were measured in length (cm) and also used to take tissue for FA analyses. Identification was to the lowest taxonomic level through macroscopic morphological analysis (MID) using identification keys [4, 27]. Moderately to advanced digested prey were identified using genetic procedures (GID) following the procedure described by [29]. Tissue samples from dorsal muscle were collected (∼10 g) and stored in ethanol (90%) at -20°C. DNA extraction was performed from 0,150 g of tissue using DNeasy® PowerSoil® Pro Kit (Cat# 47016, QIAGEN, Hilden, Germany) following the manufacturer's protocol. Each sample was quantified using

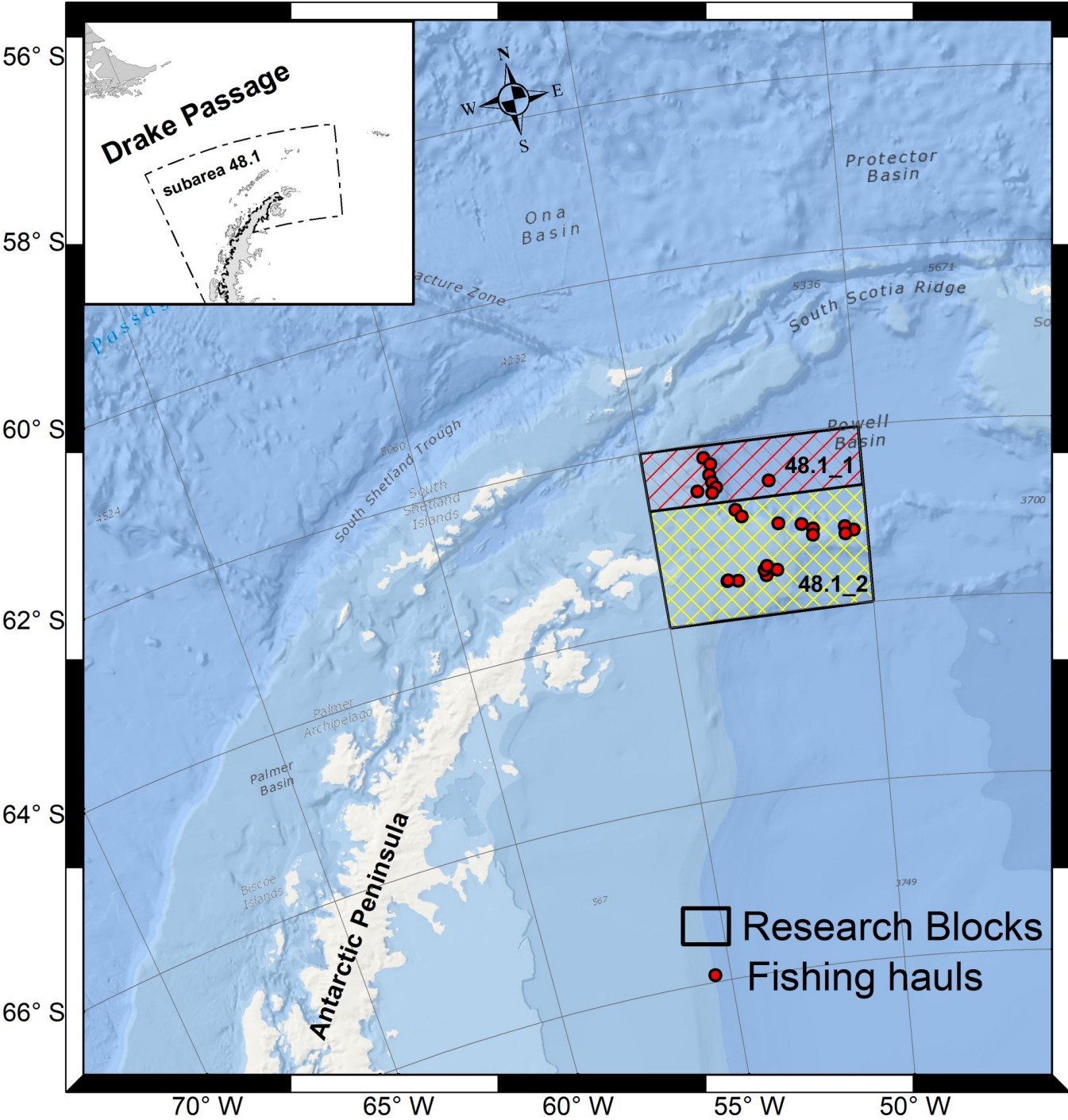

**Fig 1. Locations of the fishing hauls carried out in the northern tip of the Antarctic Peninsula (Subarea 48.1).** Boxes show CCAMLR research blocks 48.1_1 (red) and 48.1_2 (yellow), explored by the Ukrainian research program during fishing seasons 2019/20 and 2020/21. Map created in ArcMap 10.8.2 (https://www.esri.com/), using the online Ocean Basemap (https://www.arcgis.com/;itemid=5ae9e138a17842688b0b79283a4353f6). Subarea 48.1 and research blocks shapefiles downloaded from the CCAMLR geographical data available on github (https://github.com/ccamlr/data/tree/main/geographical_data).

NanoQuant microplate Infinite M200pro. To amplify the mitochondrial 16s gene, primers were synthesized and purified by HPLC at Macrogen (Korea), using the sequence described by [30] (forward: 5'–CGAGAAGACCCTRTGGAGCT–3' and reverse: 5'–GGATWGCGCTGT

TATCCCT–3′). PCR was performed using Invitrogen™ Platinum™ SuperFi™ II DNA Polymerase and the following thermocycling cycle: initial denaturation at 98˚C for 5 min, 35 cycles of initial denaturation at 98˚C for 10 s, annealing at 56˚C for 5 s, extension at 72˚C for 45 s, and a final extension at 72˚C for 5 min. Each PCR product was run on a TBE agarose gel at 1.8%, and then purified using the UltraClean® 15 DNA Purification Kit (Cat# 12100–300, MO BIO Laboratories, Inc.), following the manufacturer's protocol. The purified PCR products were quantified, and 10 ng of DNA were sent to sequencing at Macrogen using forward and reverse primer. The sequences obtained were joined and analyzed using the CLC Main Workbench (version 8.0). Taxonomic identification was performed by using BLAST with a 99% degree of similarity between the obtained sequences and reference sequences in the NCBI database.

For FA analysis, 10 g of prey muscle tissue was stored in 15 ml falcon tubes, frozen at -20˚ C and lyophilized. Samples were analyzed at the Nutrition Laboratory of the Department of Agricultural and Aquaculture Sciences of the Universidad de Magallanes (Punta Arenas, Chile). FA composition of prey was carried out by transforming them into methyl esters (FAMES) and subjected to gas chromatographic analysis, following the methodology proposed by [31], and modified by [32]. 40 mg of lyophilized sample was transferred into a test tube containing 2 mL of freshly prepared transesterification reagent (methanol: acetyl chloride, 20:1 v/v) along with 1 mL of hexane. The tubes were heated at 100˚C for 10 min in order to obtain a single methanol/hexane phase. After cooling the tubes to room temperature, 2 ml of distilled water was added to separate the mixture into two immiscible layers. The upper hexane layer (containing the methyl esters) was transferred by pipette to a chromatography vial, dried with a nitrogen stream, and finally resuspended with 300 μl of hexane.

Gas chromatography analysis was conducted using an Agilent 7890B chromatograph equipped with an autosampler and FID detector was used. In addition, a HP-88 high polarity column (60 m x 0.20 μm x 0.25 mm) was used. The carrier gas ($H_2$) flow rate was 1 ml/min and the flow split injection system with a 50:1 vent ratio was used. The injector temperature was 250˚C and the detector temperature was 280˚C. The oven temperature program was 120˚C for 5 min, with a ramp of 3˚C/min up to 220˚C (5 min). The injection volume was 1 μl and a blank was performed for every other analysis. All transesterification was performed in duplicate. Abbreviated notations of form A:B (n-x) were used, where A represents the number of carbon atoms, B represents the number of double bonds, and x gives the position of the first double bond counting from the terminal methyl group. The concentration of each fatty acid was expressed as the relative percentage of the total fatty acid content (% FAs) ± standard deviation.

## Indicators of diet composition

The diet was analyzed only in samples with stomach contents, excluding empty stomachs. Diet composition and the importance of each prey item were calculated according to the percentage of frequency of occurrence (F%), weight (W%) and number (N%), respectively expressed as:

$$F\% = \frac{Ai}{A} \cdot 100 \quad W\% = \frac{Wi}{Wt} \cdot 100 \quad N\% = \frac{Ni}{Nt} \cdot 100$$

$Ai$ is the number of individuals that consumed prey $i$, and A is the total number of stomachs examined. $Wi$ corresponds to the total wet weight of the prey $i$, and $Wt$ is the total wet weight of the prey. $Ni$ is the total number of prey $i$, and $Nt$ is the total number of prey.

From these indicators, the importance of each type of prey within the diet was calculated using the index of relative importance (IRI) [33], expressed as:

$$IRI = (W\% + N\%) \cdot F\%$$

For a better interpretation of the relative contribution of each dam, we calculated the relative importance index as a percentage (IRI%) [34], expressed as:

$$IRI\% = \frac{IRI_t}{\sum_i^n IIR} \cdot 100$$

In order to analyze diet among predator length, the IRI% was calculated into three size-class groups, G1: <100 cm, G2: 100–140 cm, and G3: 140 cm.

Feeding intensity was evaluated according to the repletion index (RI) [35], expressed as:

$$RI = \frac{mST}{mDM} \cdot 100$$

Where $mST$ is the weight of the stomach contents, and $mDM$ is the total weight of the individual.

## Statistical analysis

A Spearman's correlation test was used to measure the predator-prey length relationship using log-transformed data.

A multivariate generalized linear model (MGLM) was conducted to test variability in the prey specific numeric abundance (N) according to sex, fishing season and size-class. MGLM provides a multivariate test for the former factors and a univariate test for each prey item. The lowest Akaike's information criterion (AIC) was selected to identify the model that best explained the amount of variation in N. MGLM was run with Poisson distribution and 999 Monte-Carlo permutations using mvabund v.4.2.1 R package [36]. Statistical differences were further analyzed by pairwise post hoc comparisons between size classes using the pairwise. comp option of the anova.manyglm function. To graphically visualize variability among factors, a non-metric multidimensional scaling ordination (nMDS) plot based on Bray-Curtis distances was run using vegan v.2.6–4 R package [37].

Differences in the repletion index were assessed between sex, fishing season, and size-class, using sqrt function in a one-way analysis of variance (ANOVA).

A permutational analysis of variance (PERMANOVA) test, based on Bray–Curtis distance matrix was used to assess for differences among FAs composition of prey composition. A similarity percentage analysis (SIMPER) was also used to identify the fatty acids contributing most to differences between prey items, and a nMDS plot was used to graphically illustrate observed patterns.

All statistical analyses were performed using Rstudio 2022.12.0+353.

## Results

TOA individuals sampled in 2019/20 ranged from 64 to 174 cm TL (mean 125.1 ± 26.6 SD), with 70.5% females and 29.5% males. In 2020/21 sizes ranged from 66 to 176 cm TL (140.4 ± 24.8 cm), with 58.6% females and 41.4% males (Fig 2). From 89 stomachs sampled in 2019/20, 75 (84.3%) had contents, with a total of 186 prey items (2.5 ± 2.1 items per stomach). The prey length ranged from 4.5 to 97 cm (Fig 3A), exhibiting a significant positive predator-prey length relationship (Spearman, ρ = 0.29, p < 0.01) (Fig 3B). From 70 stomachs collected

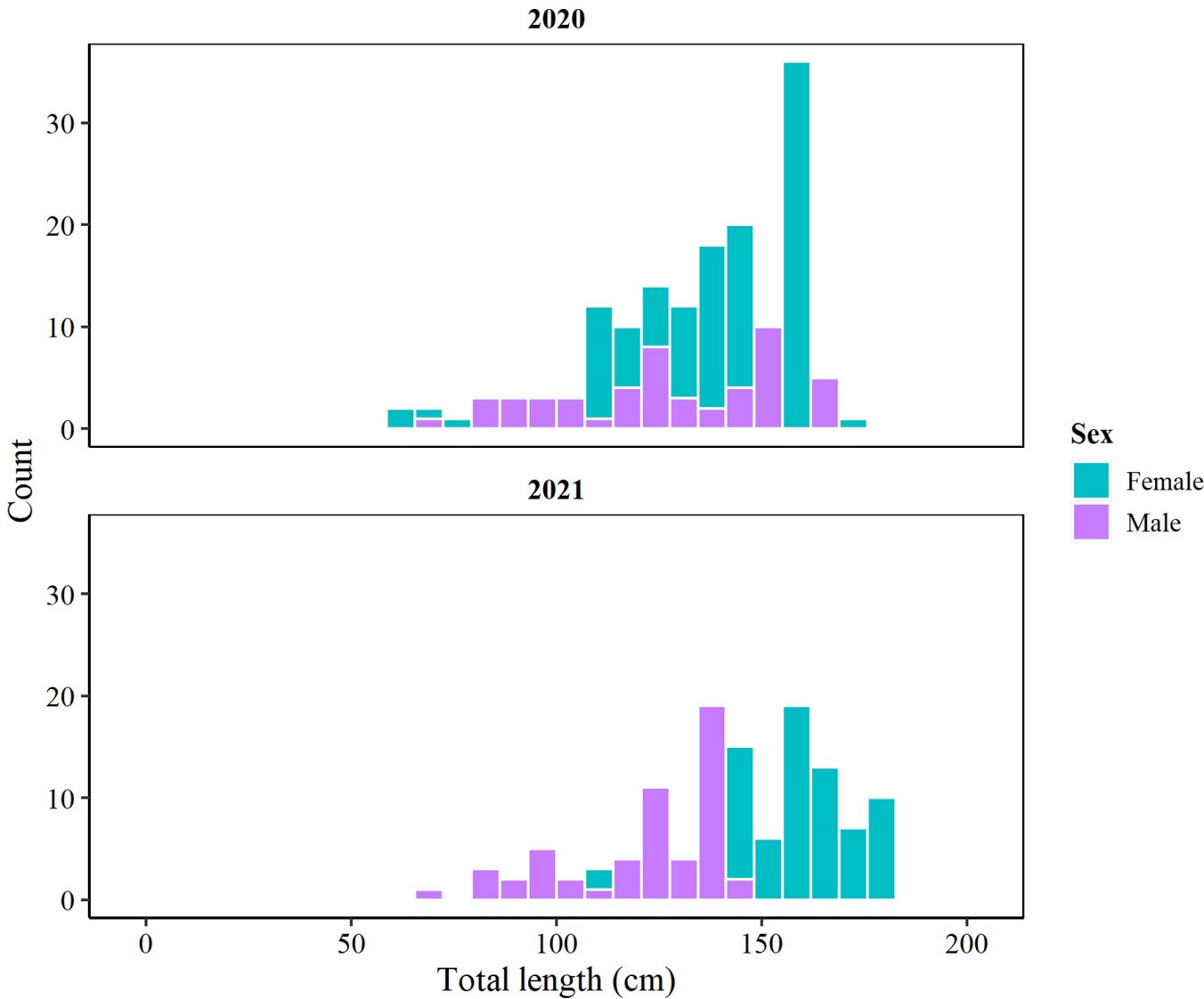

**Fig 2. Size structure of *Dissostichus mawsoni* individuals caught in the northern tip of the Antarctic Peninsula (Subarea 48.1) during the fishing seasons 2019/20 and 2020/21.**

in 2020/21, 58 (82.8%) had contents, with a total of 140 prey items (2.4 ± 2.0 prey items per stomach). The prey length ranged from 5 to 67 cm (Fig 3C), with no significant predator-prey length relationship ($\rho = 0.06$, $p = 0.69$) (Fig 3D).

In 2019/20, 4.9% of prey items were not digested, 14.2% slightly digested, 22.9% moderately digested, 39.8% advanced digested and 18.0% highly digested. In 2020/21, 5.5% of prey were not digested, 21.1% slightly digested, 27.5% moderately digested, 33.0% advanced digested, and 12.8% highly digested (S1 Fig).

## Diet composition

Fifteen taxonomic groups were recorded from stomach contents. A total of 200 prey items were identified (168 mid, 32 gid) including fishes (*Anotopterus pharao*, *Chionobathyscus dewitti*, *Macrourus whitsoni*, *Muranolepis orangensis*, *Chionodraco rastrospinosus*,

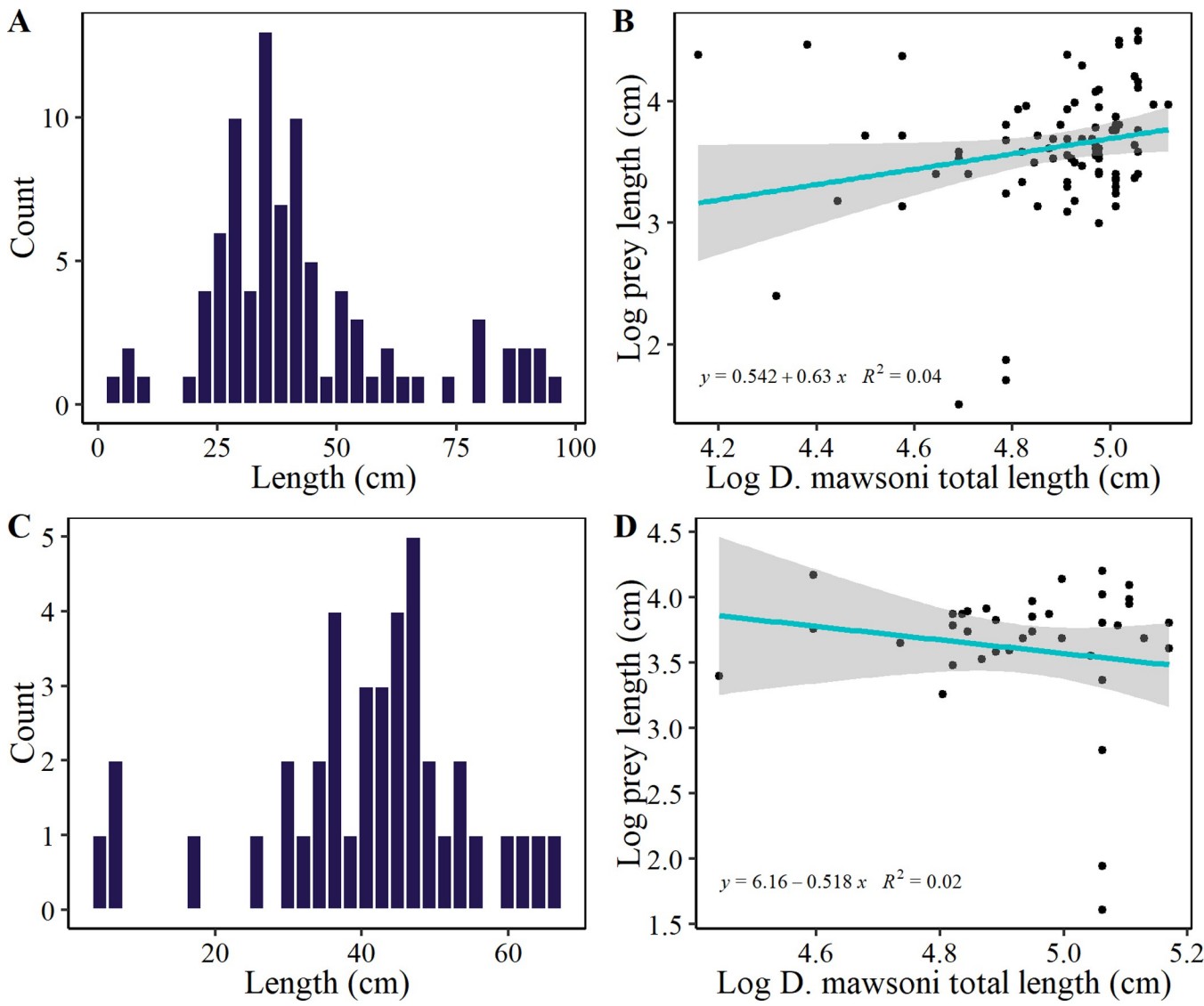

**Fig 3.** Size structure of prey items preyed by *Dissostichus mawsoni* individuals caught in the northern tip of the Antarctic Peninsula during fishing season 2019/20 (A) and 2020/21 (C). Predator-prey length relationship in both seasons (B-D) was analyzed by using a Spearman correlation test.

*Neopagetopsis ionah*, *Notothenia coriiceps*, *Trematomus eulepidotus*, *Lepidonotothen squami-frons*), crustaceans (*Eurythenes gryllus*, *Nematocarcinus lanceopes*), anthozoa, cephalopods, skates Rajidae and birds Spheniscidae (Table 1).

The TOA diet was dominated by fishes (91.0% in 2019/20, 86.3% in 2020/21) and secondarily by cephalopods (7.8% in 2019/20, 13.1% in 2020/21). Unidentified fish in advanced digestion accounted for 48.8% and 60.3% IRI for each fishing season, respectively. Among the identified fish species, Macrouridae was the most important item (32.4% in 2019/20, 25.1% in 2020/21), followed by Anotopteridae (7.4%), Channichthyidae (3.0% in 2019/20, 0.4% in 2020/21) and Nototheniidae (0.1% in 2019/20, 0.03% in 2020/21). Crustacea, Rajidae, and penguin remain reached up 0.15%, 0.06%, and 0.03% IRI, respectively (Table 1).

The MGLM showed that the prey specific numeric abundance was best explained by the additive (non-interaction) effect, where size-class and fishing season showed significant effects

**Table 1. Diet composition of *Dissostichus mawsoni* in the northern tip of the Antarctic Peninsula (Subarea 48.1) during seasons 2019/20 and 2020/21, according to percentage frequency of occurrence (F%), weight percentage (W%), number percentage (N%) and the index of relative importance (IRI%).** Parenthesis shows the number of prey items identified by macroscopic identification (MID) and genetic identification (GID).

| Prey | 2019/20 | | | | 2020/21 | | | |
|---|---|---|---|---|---|---|---|---|
| | F% | N% | W% | IRI% | F% | N% | W% | IRI% |
| Anthozoa *(MID = 1)* | - | - | - | - | 1.72 | 0.71 | 0.01 | 0.02 |
| **Mollusca** | 26.6 | 12.90 | 4.45 | 7.88 | 27.59 | 20.00 | 13.08 | 13.17 |
| Cephalopoda *(MID = 52)* | 26.67 | 12.90 | 4.45 | 7.88 | 27.59 | 20.00 | 13.08 | 13.17 |
| **Crustacea** | 5.33 | 3.23 | 0.11 | 0.15 | 6.90 | 4.29 | 0.16 | 0.15 |
| *Eurythenes gryllus (MID = 3)* | - | - | - | - | 1.72 | 2.14 | 0.10 | 0.06 |
| *Nematocarcinus lanceops (MID = 6)* | 2.67 | 2.15 | 0.09 | 0.10 | 3.45 | 1.43 | 0.04 | 0.07 |
| Unidentified crustacea | 2.67 | 1.08 | 0.02 | 0.05 | 1.72 | 0.71 | 0.02 | 0.02 |
| **Pisces** | 90.67 | 83.33 | 95.16 | 91.95 | 93.10 | 72.14 | 84.54 | 86.33 |
| Anotopteridae | 26.67 | 15.59 | 4.91 | 7.39 | - | - | - | - |
| *Anotopterus pharao (MID = 26, GID = 03)* | 2.67 | 1.61 | 1.09 | 0.12 | - | - | - | - |
| Unidentified Anotopteridae | 24.00 | 13.98 | 3.81 | 7.27 | - | - | - | - |
| Channichthyidae *(MID = 17, GID = 5)* | 13.33 | 6.45 | 13.67 | 3.06 | 8.62 | 3.57 | 5.60 | 0.41 |
| *Chionobathyscus dewitti (GID = 2)* | - | - | - | - | 3.45 | 1.43 | 1.88 | 0.16 |
| *Chionodraco rastrospinosus (GID = 1)* | 1.33 | 0.54 | 1.65 | 0.05 | - | - | - | - |
| *Neopagetopsis ionah (GID = 2)* | 1.33 | 0.54 | 1.01 | 0.04 | 1.72 | 0.71 | 1.15 | 0.05 |
| Unidentified Channichthyidae | 10.67 | 5.38 | 11.02 | 2.98 | 3.45 | 1.43 | 2.56 | 0.20 |
| Macrouridae | 48.00 | 23.66 | 54.62 | 32.46 | 43.10 | 22.14 | 53.22 | 25.13 |
| *Macrourus whitsoni (MID = 17, GID = 19)* | 17.33 | 8.06 | 28.98 | 10.93 | 25.86 | 15.00 | 36.36 | 19.16 |
| Unidentified Macrouridae | 30.67 | 15.59 | 25.63 | 21.52 | 17.24 | 7.14 | 16.86 | 5.97 |
| Muraenolepididae | 1.33 | 0.54 | 0.54 | 0.02 | 5.17 | 2.14 | 3.34 | 0.41 |
| *Muranolepis orangensis (GID = 4)* | 1.33 | 0.54 | 0.54 | 0.02 | 5.17 | 2.14 | 3.34 | 0.41 |
| Nototheniidae | 2.67 | 2.69 | 4.38 | 0.16 | 1.72 | 0.71 | 0.38 | 0.03 |
| *Lepidonotothen squamifron*s (MID = 4) | 1.33 | 2.15 | 3.56 | 0.13 | - | - | - | - |
| *Nothotenia coriiceps (GID = 1)* | 1.33 | 0.54 | 0.82 | 0.03 | - | - | - | - |
| *Trematomus eulepidotus (GID = 1)* | - | - | - | - | 1.72 | 0.71 | 0.38 | 0.03 |
| Rajidae | 1.33 | 0.54 | 2.06 | 0.06 | - | - | - | - |
| Unidentified Rajidae | 1.33 | 0.54 | 2.06 | 0.06 | - | - | - | - |
| Unidentified pisces | 58.67 | 33.87 | 14.98 | 48.80 | 63.79 | 43.57 | 22.01 | 60.35 |
| **Bird** | - | - | - | - | 5.17 | 2.14 | 2.07 | 0.31 |
| Spheniscidae *(MID = 3)* | - | - | - | - | 5.17 | 2.14 | 2.07 | 0.31 |
| **Other** | 1.33 | 0.54 | 0.28 | 0.02 | 1.72 | 0.71 | 0.13 | 0.02 |
| Unidentified | 1.33 | 0.54 | 0.28 | 0.02 | 1.72 | 0.71 | 0.13 | 0.02 |

(Table 2). The nMDS plot showed that prey specific abundance was different between G2 and G3 size groups than for G1, showing differences in diet between size groups (Fig 4). The observed differences were confirmed by MGLM analyses indicating that the diet composition differed significantly among size-class and fishing season (Table 2). Pairwise comparisons showed that diet differed mainly between G2 and G3 (S2 Table). The variable sex was discarded during the AIC selection process, in order to obtain the most parsimonious approach (that is, to select a model that better explains our data using fewer parameters) (S3 Table).

Among identified prey, a significant effect of size-class was recorded on cephalopod consumption (Table 2) that was slightly present in groups G2 and G3 in 2019/20, being more relevant in 2020/21, especially G1 exceeding 55.0% IRI (Fig 5). On the other hand, a significant effect of the fishing season was found on Anotopteridae consumption (Table 2), which

**Table 2. Results of MGLM testing effect of size-class and fishing season on the prey specific numeric abundance in the stomachs of the Antarctic toothfish from the northern tip of the Antarctic Peninsula during fishing seasons 2019/20 and 2020/21.** Ceph = Cephalopods, Ano = Anotopteridae, Cha = Channichthyidae, Mac = Macrouridae, Raj = Rajidae, Not = Nototheniidae, Mur = Muranoloepididae, Sph = Spheniscidae, Cru = Crustacea.

| | Multivariate | | | | Univariate | | | | | | | | |
|---|---|---|---|---|---|---|---|---|---|---|---|---|---|
| Factors | Res.Df | Df.diff | Dev | p (>Dev) | Cep | Ano | Cha | Mac | Raj | Not | Mur | Sph | Cru |
| (Intercept) | 132 | | | | | | | | | | | | |
| Size class | 130 | 2 | 59.31 | **0.01*** | **0.01*** | 0.94 | 0.88 | 0.55 | 0.94 | 0.17 | 0.94 | 0.94 | 0.88 |
| Fishing season | 129 | 1 | 48.35 | **0.00*** | 0.87 | **0.00*** | 0.66 | 0.87 | 0.86 | 0.60 | 0.69 | 0.39 | 0.87 |

dominated G1 in 2019/20 (67.65% IRI) and was absent in 2020/21 (Fig 5). Although no significant effect of size group on other species was found, it can be seen that Macrouridae was less present in group G1 (29.99% IIR in 2019/20), but dominated the diet of groups G2 and G3

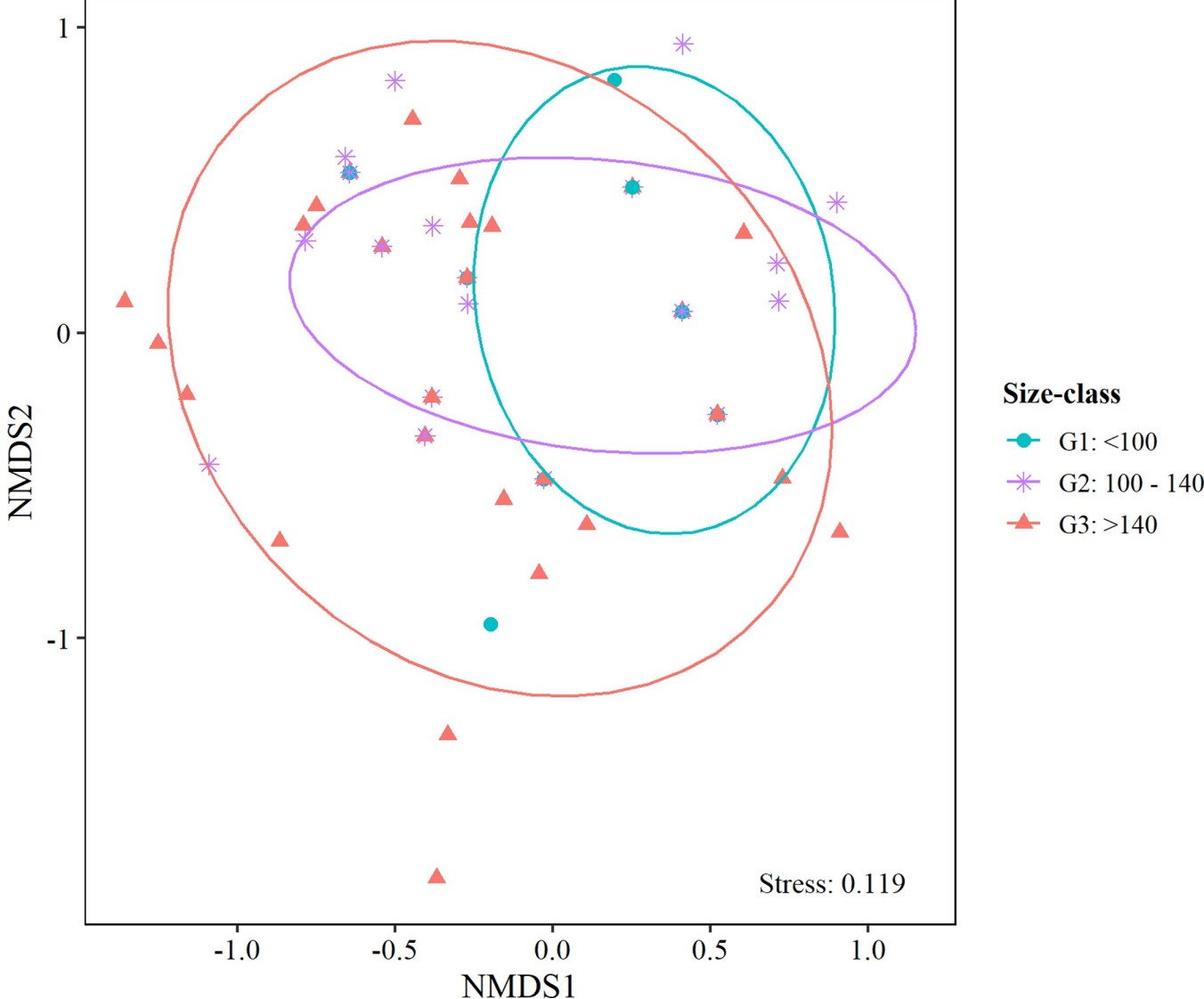

**Fig 4.** Nonmetric multidimensional scaling (nMDS) plot of prey specific numeric abundance among size-classes (G1: <100 cm (n = 16); G2: 100–140 cm (n = 64); G3: >140 cm (n = 43)). Ordination based on Bray-Curtis distance matrix.

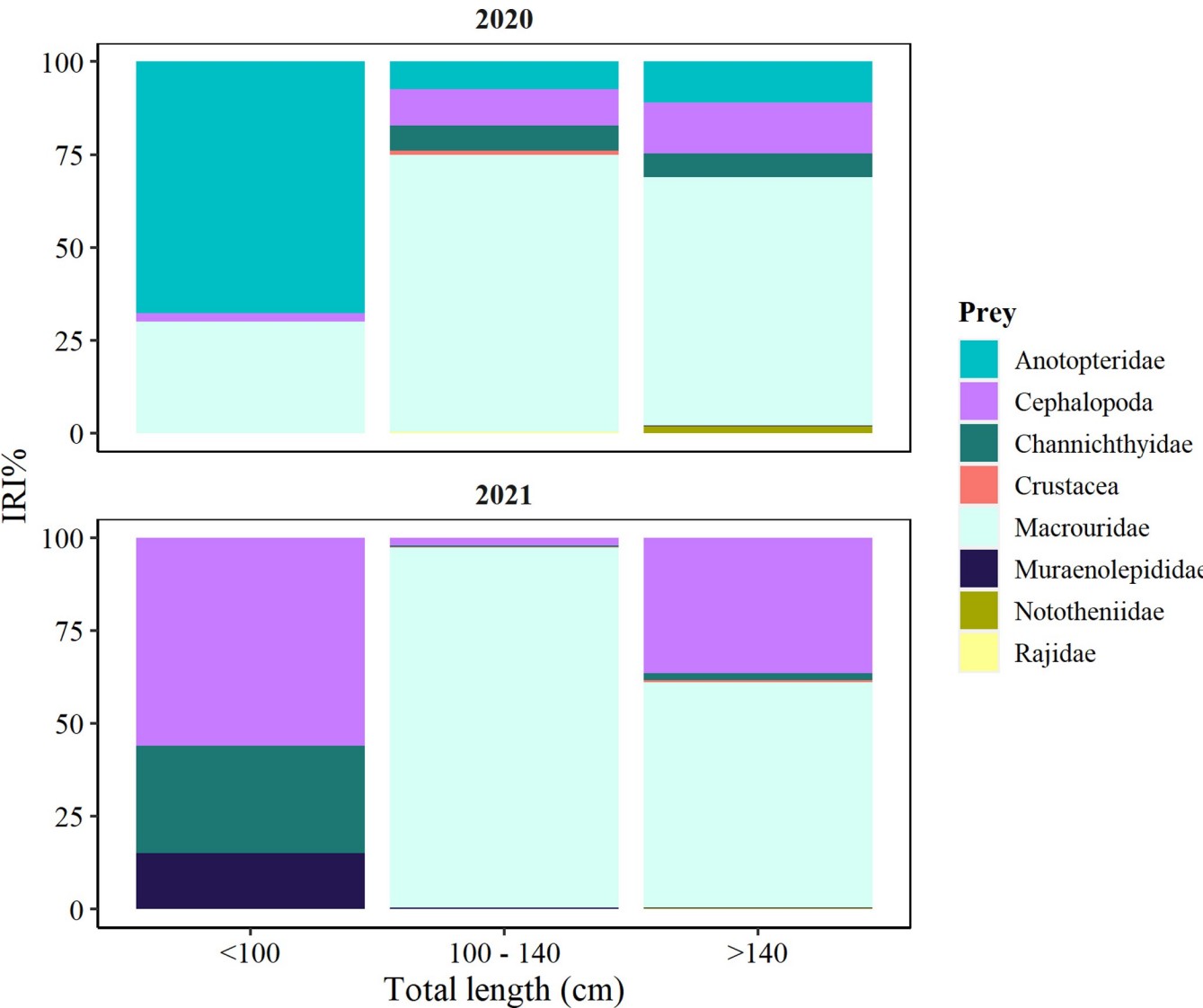

**Fig 5.** Diet composition of *Dissostichus mawsoni* according to size classes G1: <100 cm (n = 16); G2: 100–140 cm (n = 64); G3: >140 cm (n = 43), collected in the northern tip of the Antarctic Peninsula (Subarea 48.1) during fishing seasons 2019/20 and 2020/21.

with 74.44% to 66.86% IRI, respectively, and Channichthyidae were particularly important to G1 in 2020/21 (28.87% IRI) (Fig 5).

Feeding intensity was low (Repletion Index = 1.23% ± 1.42 SD) with no significant differences between fishing season (ANOVA, $F_{(1, 154)}$ = 0.0004 p = 0.994), sex (ANOVA, $F_{(1, 154)}$ = 0.057, p = 0.812) and size classes (ANOVA, $F_{(2, 153)}$ = 0.102 p = 0.902).

## Fatty acid composition of prey items

A total of 29 fatty acids were identified for Anotopteridae, Channichthyidae, Macrouridae and Cephalopoda. At the species level, we identified profiles for *Chionobathyscus dewitti* and *Macrourus whitsoni* (Table 3). Overall, prey items were rich in PUFA acids (ranging from 39.8% for Macrouridae to 49.3% for cephalopods), with a high contribution of docosexanoic acid (C22:6 n-3) and eicosapentaenoic (C20:5 n-3). Secondarily, MUFA acids (ranging from 24.5% for

**Table 3. Fatty acid (FA) composition (% of total FA, mean ± sd) of prey items in the *Dissostichus mawsoni* diet from individuals collected in the northern tip of the Antarctic Peninsula.** (SAFA are saturated fatty acids, MUFA are monounsaturated fatty acids and PUFA are polyunsaturated fatty acids).

| Fatty acid | Macrouridae n = 9 | | M. whitsoni n = 5 | | Anotopteridae n = 1 | Channichthyidae n = 2 | | Cephalopoda n = 11 | |
|---|---|---|---|---|---|---|---|---|---|
| | mean | sd | mean | sd | | mean | sd | mean | sd |
| C14:0 | 1.91 | 0.46 | 1.63 | 0.47 | 1.90 | 3.62 | 4.19 | 2.48 | 0.82 |
| C15:0 | 0.02 | 0.07 | 0.00 | 0.00 | 0.00 | 0.16 | 0.23 | 0.01 | 0.03 |
| C16:0 | 16.83 | 3.14 | 18.59 | 1.46 | 18.10 | 15.82 | 1.28 | 19.06 | 2.56 |
| C17:0 | 0.00 | 0.00 | 0.00 | 0.00 | 0.00 | 0.00 | 0.00 | 0.02 | 0.07 |
| C18:0 | 3.77 | 0.92 | 4.62 | 0.38 | 3.70 | 3.07 | 2.21 | 3.15 | 0.95 |
| C21:0 | 0.19 | 0.38 | 0.00 | 0.00 | 0.00 | 0.00 | 0.00 | 0.08 | 0.27 |
| *Σ SAFA* | *22.73* | *3.22* | *24.84* | *1.71* | *23.70* | *22.68* | *0.93* | *24.80* | *2.15* |
| C14:1 | 0.02 | 0.07 | 0.00 | 0.00 | 0.00 | 0.25 | 0.36 | 0.00 | 0.00 |
| C16:1n7 | 5.77 | 2.09 | 4.09 | 1.26 | 3.60 | 6.68 | 6.11 | 1.92 | 1.33 |
| C17:1 | 0.00 | 0.00 | 0.00 | 0.00 | 0.00 | 0.00 | 0.00 | 0.03 | 0.09 |
| C18:1n9 | 0.00 | 0.00 | 0.00 | 0.00 | 0.00 | 0.00 | 0.00 | 0.01 | 0.03 |
| C18:1n9cis | 16.70 | 4.35 | 13.87 | 4.03 | 12.80 | 14.03 | 8.61 | 5.44 | 2.61 |
| C18:1n7 | 5.68 | 0.93 | 4.50 | 1.02 | 8.90 | 7.16 | 0.20 | 4.51 | 0.63 |
| C20:1 n9 | 4.04 | 2.29 | 3.54 | 1.64 | 2.20 | 1.73 | 0.16 | 9.11 | 1.11 |
| C22:1n9 | 1.93 | 0.86 | 2.06 | 0.59 | 2.30 | 2.89 | 3.09 | 3.22 | 1.17 |
| C24:1n9 | 1.10 | 0.32 | 1.46 | 0.17 | 1.00 | 1.03 | 0.89 | 0.27 | 0.30 |
| *Σ MUFA* | *35.23* | *8.45* | *29.52* | *6.28* | *30.70* | *33.76* | *10.73* | *24.50* | *5.09* |
| C16:3 n4 | 0.06 | 0.12 | 0.00 | 0.00 | 0.00 | 0.11 | 0.15 | 0.01 | 0.03 |
| C18:2n6 | 0.00 | 0.00 | 0.00 | 0.00 | 0.00 | 0.17 | 0.25 | 0.00 | 0.00 |
| C18:2n6cis | 1.38 | 0.21 | 1.21 | 0.19 | 1.30 | 2.04 | 0.48 | 0.14 | 0.28 |
| C18:3n3 | 0.12 | 0.23 | 0.00 | 0.00 | 0.00 | 0.55 | 0.78 | 0.00 | 0.00 |
| C18:4n3 | 0.84 | 0.42 | 0.59 | 0.53 | 0.00 | 1.31 | 1.86 | 0.09 | 0.22 |
| C18:3n6 | 0.01 | 0.03 | 0.00 | 0.00 | 0.00 | 0.00 | 0.00 | 0.00 | 0.00 |
| C20:2n6 | 0.03 | 0.10 | 0.00 | 0.00 | 0.00 | 0.00 | 0.00 | 0.91 | 0.67 |
| C20:3n6 | 0.01 | 0.03 | 0.00 | 0.00 | 0.00 | 0.00 | 0.00 | 0.00 | 0.00 |
| C20:3n3 | 0.00 | 0.00 | 0.26 | 0.59 | 0.00 | 0.00 | 0.00 | 0.12 | 0.20 |
| C20:4n6 | 1.04 | 0.79 | 0.59 | 0.46 | 0.00 | 0.44 | 0.23 | 1.69 | 1.16 |
| C20:4n3 | 0.12 | 0.24 | 0.11 | 0.15 | 0.30 | 0.19 | 0.27 | 0.00 | 0.00 |
| C20:5n3 | 14.22 | 4.00 | 14.93 | 1.63 | 13.80 | 15.65 | 4.44 | 16.92 | 2.03 |
| C22:5n3 | 0.98 | 0.23 | 1.30 | 0.34 | 1.50 | 0.64 | 0.07 | 0.98 | 0.57 |
| C22:6n3 | 21.03 | 6.01 | 25.99 | 4.19 | 28.40 | 20.34 | 12.82 | 28.51 | 7.89 |
| *Σ PUFA* | *39.84* | *7.88* | *44.97* | *5.24* | *45.30* | *41.45* | *14.27* | *49.36* | *5.29* |
| *Σ FAs* | *97.78* | *2.61* | *99.31* | *1.09* | *99.70* | *97.90* | *2.60* | *98.65* | *1.50* |
| Σ ω3 | 37.31 | 8.73 | 43.18 | 5.79 | 44.00 | 38.69 | 14.42 | 46.61 | 5.84 |
| Σ ω6 | 2.48 | 1.01 | 1.80 | 0.60 | 1.30 | 2.66 | 0.00 | 2.74 | 0.88 |
| *ω3/ω6* | *15.06* | *8.65* | *27.34* | *12.78* | *33.85* | *14.56* | *5.42* | *19.08* | *7.79* |

cephalopods to 35.2% for Macrouridae) with oleic acid (C18:1 n-9) being higher in fishes (10.8% for Anotopteridae and 16.7% for Macrouridae) than cephalopods (5.4%), and conversely the eicosanoic acid (C20:1 n-9) was higher in cephalopods (9.1%) than fishes (2.2% and 4.0% for Anotopteridae and Macrouridae, respectively). Thirdly, SAFA acids (ranging from 22.7% for Macrouridae to 24.8% for cephalopods), dominated by palmític acid (C16:0) of similar values for Macrouridae (16.8%), Anotopteridae (18.1%) and cephalopods (19.0%). The nMDS analysis revealed two prey groups of fishes (Anotopteridae-Macrouridae-Channichthyidae) and

Cephalopoda (Fig 6), that were statistically different (F(4, 20) = 4.58; p<0.001). According to the SIMPER analysis, the fatty acids that contributed most to the dissimilarities between groups were oleic C18:1 n9cis, eicosanoic C20:1 n9, docosexanoic C22:6n3 DHA, linoleic C18:2 n6cis, palmitoleic C16:1n7 and octatetradenoic C18:4n3 (S4 Table).

## Discussion

The present study provides the first comprehensive description of the feeding behavior of the Antarctic toothfish by combining stomach content identification with fatty acid analysis, providing a wider understanding of the feeding ecology and role of TOA in the Antarctic Peninsula. Trophic dynamics is a prerequisite for ecosystem-based fisheries management and necessary to assess potential fishing impacts on target species, which is especially relevamt in the Antarctic Peninsula, where direct fishing for notothenioids has been prohibited.

### Feeding behavior

In the Antarctic Peninsula, TOA feeds mainly on fishes, a prey of high nutritional value and often the most available, being usually the most important prey item for top predators in Antarctica [27, 38]. From these fishes, Macrouridae was the most important prey, similar to what has been described in previous studies from other regions, where *Macrourus whitsoni* and *Macrourus caml*, a sympatric species inhabiting the same depth range with TOA (900 and 1900 m depth), frequently appears in TOA diet and as bycatch in the fishery [5, 8, 16, 23, 27, 39–41]. Considering that the TOA diet is dominated by locally abundant fish species [5, 42], and based on the high contribution of Macrouridae to the TOA diet observed in this study, we can assume that this group is probably the most abundant fish in the AP. Although there are no direct biomass estimations for Macrouridae in the AP, previous evidence has shown that this group is the most widespread representative of bycatch in 2019/20 and 2020/21 [43].

Cephalopods were reported as an important prey item [8, 5], whereas Channichthyidae was an important prey among fishes. Within this group, we identified *Chionobathyscus dewitti*, a species reported as one of the main prey of TOA in the Lazarev Sea and the Ross Sea [5, 39], which is commonly found between 600 to 1600 m depth in the Antarctic Peninsula [44, 45], and over 2000 m depth in the Weddell Sea [45, 46]. From the fishery data, Channichthyidae is the second most abundant bycatch group in the Antarctic Peninsula [43].

Surprisingly, we observed a low contribution of Nototheniidae, even in individuals <100 cm, compared to the diet of individuals from East Antarctica where it can reach up to 35% [8]. It is possible that for greater energy efficiency, individuals prey on a smaller number of larger and heavier Macrouridae individuals, rather than a larger number of smaller and lighter Nototheniidae [8]. Another explanation could be that the fish community in the AP is heavily dominated by Macrouridae, with reduced availability of Nototheniidae as a consequence of the population collapse in past decades [15]. Although, there was an important amount of highly digested unidentified fish prey (which reached up to 60% IRI) where Nototheniidae could be present. In addition, reduced presence of Notheniidae was also observed in the fishery with low bycatch estimates in the area [43].

Benthic crustaceans such as *Nematocarcinus lanceopes* and *Eurythenes gryllus* were also found. Both species are relatively rare and have been recorded at depths between 500 and 2031 m for *N. lanceopes* [47, 48], whereas *E. gryllus* has been reported between 550 to 7800 m depth [49, 50], overlapping with TOA depth range. Another less important item was penguin remains (found in two stomachs), suggesting that part of the diet may have come from carrion. Also, we found coral and rock fragments, which is indicative of benthic foraging habits [8, 16, 27].

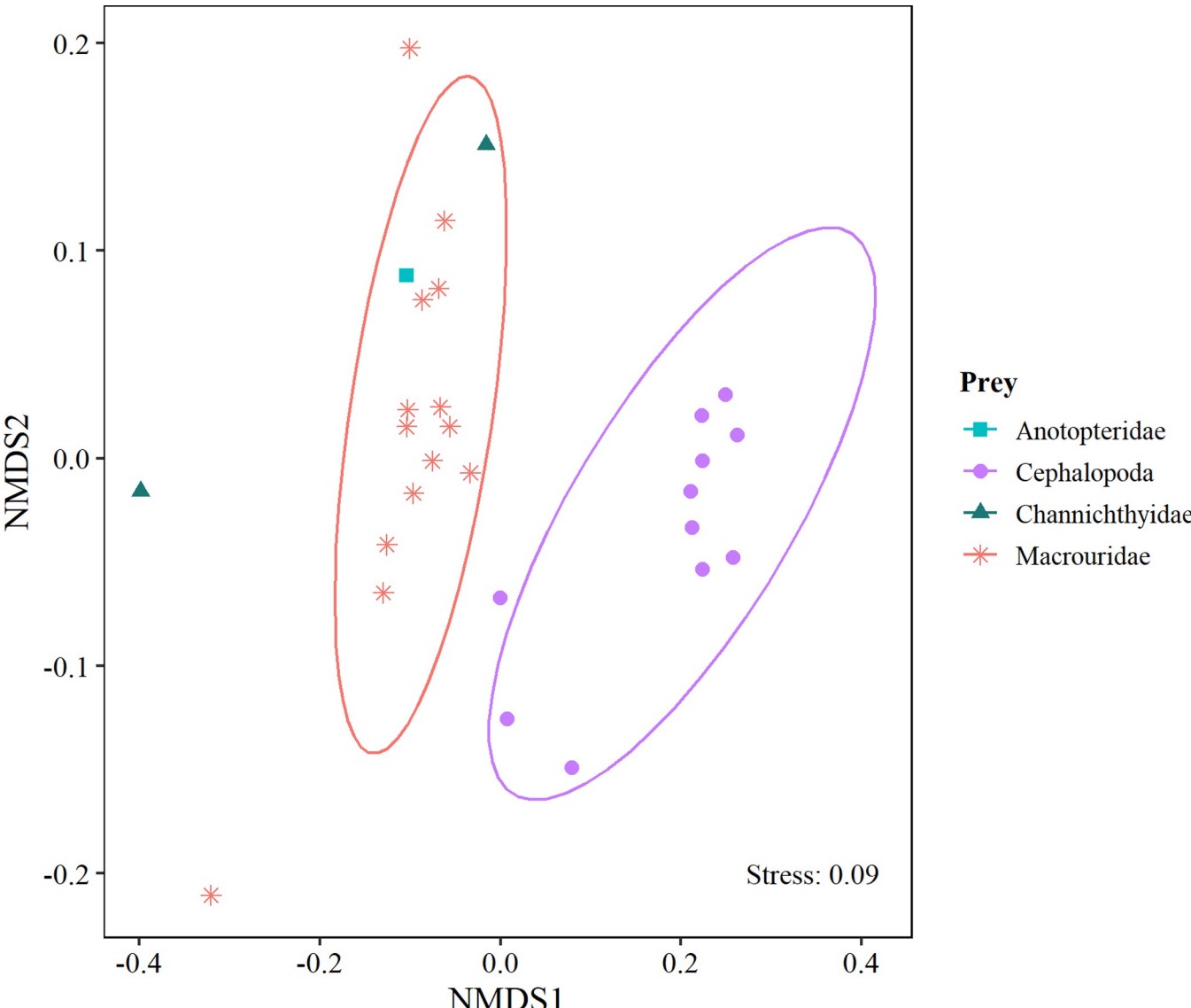

**Fig 6. Nonmetric multidimensional scaling (nMDS) plot of fatty acid data (expressed as the percentage of total fatty acids, %FAs) from the four identified prey groups recorded in stomachs of *Dissostichus mawsoni*.** Ordination based on Bray-Curtis distance matrix.

The former prey composition is representative of benthopelagic fauna [4, 45], confirming TOA as a demersal feeder whose feeding preferences are restricted to the abundance of different prey [5, 8, 16]. Also, the presence of *Anotopterus vorax* has previously been associated with migration to shallow waters [27]. This information confirms that the TOA is a top predator capable of impacting and controlling the bentho-pelagic ecosystem in the Antarctic Peninsula.

A positive correlation was observed between TOA size and prey size, with prey size increasing as predator size increased. This is common as predator-prey interactions often depend on body size, due to the morphological limitations of the predator in agreement with the morphological ontogenetic changes in the feeding apparatus [51, 52]. In other diet studies, TOA subadults showed a diet composed of a variety of smaller prey, such as smaller fishes and benthic crustaceans, while adult individuals preyed mostly on larger demersal fishes, such as Macrouridae [16]. One point of variability was the weak predator-prey relationship as illustrated in

Fig 3B and 3D, resulting from the presence of Anotopteridae, which, although exaggerated in length, can generally be considered small prey, and therefore consumed by small TOA individuals.

Significant differences in diet composition were found between size groups, with dietary variability being greater in adults (>100 cm) than in juveniles (<100 cm). This can be related to changes in buoyancy and depth range distribution [28]. It is known that TOA juveniles have a negative buoyancy, constraining depth range to shallower waters and benthic habitats, while adults have neutral buoyancy, being capable to move across a wider depth range from shallow to deep waters [4, 6, 16, 28]. This adaptation would allow TOA to avoid predators and generate changes in accessibility to different prey items. Some prey, such as Cephalopoda and Channichthyidae, have benthic-pelagic habits [45, 53], thus being important for juveniles. On the other hand, prey such as Macrouridae are abundant in deep waters [40], becoming more important for TOA adults. Other studies have also found changes in diet with predator length [5, 8, 17, 54], which could demonstrate that there are ontogenetic changes in the diet of Antarctic toothfish. However, because all fishes in this study were caught deeper than 900 m depth, juveniles were not represented and the smallest individuals analyzed were 80 cm; hence, it is necessary to extend to smaller sizes probably on shallow waters to obtain more determinant results.

Other sources of variability in TOA diet have been attributed to different areas and depth strata [8, 21], but interannual variability is rarely analyzed. Considering that the TOA diet can strongly reflect local fish assemblages [42], it is interesting the changes in prey composition between two consecutive fishing seasons observed in this study. In Antarctica, demersal fish assemblages can vary according to water temperature [55], however this was not assessed in this study. The ongoing oceanographic changes well-described along the Antarctic Peninsula [56], could also be impacting the deep-water fish assemblage; hence it is important to measure physical factors to assess if they are driving changes in the distribution of prey in the area.

## Fatty acids prey composition

Overall, PUFA was twice as numerous as MUFA and SAFA in all analyzed prey, which is common in high-latitude ecosystems [57]. SAFA and MUFA are used as metabolic energy in the lipoproteins synthesis [58], whereas PUFAs are incorporated into lipoproteins [59], being used to synthesize the tissues of the developing organism [60]. PUFAS are fatty acids with more than two double bonds and 18 carbons in their chemical constitution, making them highly nutritious. We also observed higher content of w3 than w6 (high w3/w6 ratio), characteristic of marine ecosystems [61], being beneficial for cell membranes of most tissues, since it is known that w3 acids are much more useful for energy expenditure than for tissue construction [60]. For example, specifically, the n-3/n-6 balance of membranes and micro domains of lipid membranes strongly influences cellular processes by modulating the expression of different genes affecting cell survival [62] and therefore can mean a higher or lower survival success.

Fish and cephalopod prey showed similar amounts of saturated fatty acids (SAFAs), mostly dominated by palmític acid (C16:0), one of the most common FAs in marine organisms, key to providing metabolic energy during growth and spawning [63]. Among MUFAs there were higher values in fishes rather than cephalopods, especially oleic acid (C18:1), a fatty acid commonly found in prey that lives at great depths [64, 65]. Both fishes and cephalopods were prey rich in PUFAs eicosapentaenoic (C20:5) and docosahexaenoic (C22:6 or DHA), often considered essentials acids as most aquatic animals are not capable of synthesizing [63]. In particular, DHA (which contributed upon 20% of FA composition in all measured prey) is indicative of optimal animal health increasing individual survival and reproductive success [63, 66]. DHA

plays a major role in the maintenance of tissue cell membranes, especially those of the retina and brain [67], and for example in marine fish larvae, its absence are related to abnormal neural and visual development [68]. According to results, TOA is therefore assimilating DHA through the diet by consuming prey of high nutritional value.

Also, TOA feeds on prey with a high proportion of MUFAs and PUFAs, where both fishes and cephalopods can supply similar amounts of energy, so there could be no need for a specialist behavior, potentially explaining the generalist strategy commonly associated with TOA feeding. Other predators tend to have a selective behavior as the differential energetic contribution among different prey [69]. Since we were not able to analyze predator tissues we could not compare the fatty acid signatures among the predator and their prey; this would have provided us with important information of quantitative trophic predator–prey relationship [70]. However, the information provided here still constitutes key information to improve our understanding of energy flow and carbon transfer pathways in relation to the role of TOA in the benthic-demersal ecosystem of the Antarctic Peninsula.

## Management consideration

Antarctic toothfish has been consistently managed by the CCAMLR as a key component in the food web dynamics of the Antarctic ecosystem [5, 22]; however, issues including the population status, the effect of environmental change on its complex life cycle, and ecosystem impacts of the fisheries are still to be well understood [71]. Moreover, climate change projections suggest that warming will negatively affect toothfish and other benthopelagic dwelling species [72]; however, the effects of climate change on TOA are still unclear as while some studies suggest this species will respond negatively to warming due to its physiology, others suggest it may be able to adapt considering its capacity to inhabit different habitats [73]. For this reason, a complete understanding of feeding dynamics and trophic connections is relevant for ecosystem-based stock management in order to develop fishing activities while maintaining ecosystem structure and functioning. One risk in the toothfish fishery is the predation release, where fishing-induced changes in predator-prey relationship may generate the unbalanced proliferation of one species, destabilizing the structure and function of the food-web [21]. This process could be of particular concern in environments already modified by overexploitation, such as the Antarctic Peninsula, where it is unknown how TOA and the surrounding bentho-demersal community will respond to fishing pressure, also considering the impacts of climate change in the area.

## Conclusion

This study described for the first time the diet of TOA in the Antarctic Peninsula, where fishes and cephalopods dominated the diet, especially Macrouridae which is probably the most abundant fish in the AP. Diet varied between size-classes suggesting a ontogenetic change in prey preferences from Anotopteridae in smaller individuals to Macrouridae in higher ones, showing preference for prey items with high nutritional value (rich on PUFA DHA), suggesting that TOA assimilates essential fatty acids through the diet in order to maintain optimal health. Results presented here are relevant for fisheries management as TOA's prey also constitutes bycatch species, which is particularly relevant for the benthic-demersal fish communities at the Antarctic Peninsula. Since some of these species were overexploited in the past, and considering nowadays it is still unknown the extent of its recovery and how TOA population is shaping by predation. As CCAMLR effort are focused on managing fish stocks in a precautionary and ecosystemic manner, and considering the uncertainty associated with the projected

climatic changes in the area and its effects on the ecosystem, more studies on predation inter-action, energy flow and structure stability of the entire food web are needed.

## Supporting information

**S1 Table. Date, haul number, geographic position and depth (m) of fishing hauls carried out by the Ukrainian commercial vessel Calipso in the Antarctic Peninsula (Subarea 48.1) during 2019/20 and 2020/21.**
(DOCX)

**S2 Table. Factors tested in MGLM to best explain the amount of variation in prey specific abundance in the stomach content of the Antarctic toothfish in the Antarctic Peninsula.** Degrees of freedom (d.f.), Log-likelihood (LL), Akaike Information Criteria (AIC) and incre-ments of each model in comparison with the best model (ΔAIC) are shown.
(DOCX)

**S3 Table. Post hoc pairwise comparisons of manyglm analyses, between size-class catego-ries.**
(DOCX)

**S4 Table. SIMPER results of fatty acid profile %FAs data with pairwise tests between the four identified prey groups.** The average dissimilarity (Av. diss, %) between groups is shown in the first row. FAs are ordered according to their percentage contribution (Contrib%) of the total dissimilarity with a 50% cutoff level.
(DOCX)

**S1 Fig. Digestion level of prey items in the stomach of *Dissostichus mawsoni* collected in the northern tip of the Antarctic Peninsula during fishing season 2019/20 (n = 186) and 2020/21 (n = 140).**
(DOCX)

**S1 Data.**
(XLSX)

## Acknowledgments

The authors thank the crew and scientific observer of the Ukrainian vessel Calipso.

## Author Contributions

**Conceptualization:** Karina Pérez-Pezoa, César A. Cárdenas, Marcelo González-Aravena, Francisco Santa Cruz.

**Data curation:** Karina Pérez-Pezoa, Pablo Gallardo, Alí Rivero, Vicente Arriagada, Francisco Santa Cruz.

**Formal analysis:** Karina Pérez-Pezoa, Francisco Santa Cruz.

**Funding acquisition:** César A. Cárdenas, Marcelo González-Aravena.

**Supervision:** César A. Cárdenas, Francisco Santa Cruz.

**Visualization:** Karina Pérez-Pezoa, Francisco Santa Cruz.

**Writing – original draft:** Karina Pérez-Pezoa, César A. Cárdenas, Marcelo González-Aravena, Pablo Gallardo, Alí Rivero, Vicente Arriagada, Kostiantyn Demianenko, Pavlo Zabroda, Francisco Santa Cruz.

**Writing – review & editing:** César A. Cárdenas, Francisco Santa Cruz.

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
