## [Decision Letter · Decision Letter 0]

19 Apr 2023

PONE-D-23-08000Trophodynamics of the Antarctic toothfish (Dissostichus mawsoni) in the Antarctic Peninsula: Ontogenetic changes in diet composition and prey fatty acid profilesPLOS ONE

Dear Dr. Francisco Santa Cruz

Thank you for submitting your manuscript to PLOS ONE. After careful consideration, we feel that it has merit but does not fully meet PLOS ONE’s publication criteria as it currently stands. Therefore, we invite you to submit a revised version of the manuscript that addresses the points raised during the review process.

ACADEMIC EDITOR: Please insert comments here and delete this placeholder text when finished. Be sure to:

 The authors need to improve the article alot in terms of language and comments given by the esteemed reviewers. So I recommends its major revision.

Reviewer #1: have checked the paper entitled " Trophodynamics of the Antarctic toothfish (Dissostichus mawsoni) in the Antarctic Peninsula: Ontogenetic changes in diet composition and prey fatty acid profiles" The paper was very good and well written but needed some corrections as below:

1- For molecular analyses, you reached the phylogenetic tree or only by BLAST, the sequences checked, and what species compared with?

2- Please improve the English language for the paper

3- In Fig. 3. R2 ratio is so weak; do you have any explanation for it?

4- The rest of the comments are implemented in the attached file.

Reviewer #2: Recent study on "Trophodynamics of the Antarctic toothfish (Dissostichus mawsoni) in the Antarctic

Peninsula: Ontogenetic changes in diet composition and prey fatty acid profiles" base on Prey- Predator relationship in aquatic ecosystem is reliable which gives the information about the food chain also. The study of feeding material or prey based on molecular basis is give the trustable impact of the effective study But in graphical representation figure no.3 between in numbers and length is confusable regarding the depending and independing varriable (length,Number) for choosing the cordinates (X and Y). so please check it and see the manuscript which is attach with minor correction in Graph.

Reviewer #3: I have evaluated the manuscript “PONE-D-23-08000 Trophodynamics of the Antarctic toothfish (Dissostichus mawsoni) in the Antarctic Peninsula: Ontogenetic changes in diet composition and prey fatty acid profiles. I am suggesting the authors and curious to know the some major observations-

1. Authors did two sampling in two successive years in the month of February. What was the reason for choosing the month February? Why other months were not chosen for sampling? Whether two samplings are enough to determine the feeding habits of different species? Explain it.

2. Authors dissected out the stomach of different samples and classified the prey items based on the extent of digestion in stomach. Digestion starts in stomach and only protein is digested in stomach. Then how can you have classified them in different extent of digestion? In this scenario, how did you take the sample of these preys? And if prey were gone digestion process, whether it represented the actual fatty composition? Authors have performed fatty acid composition and correlated with nutritional quality of prey. What was the reason behind it? How can authors correlates the fatty acid composition with prey’s nutritional quality.

3. The manuscript is well written and results are well described. Data were well statistically analysed. Results are properly discussed in discussion section. The authors need to revise manuscript in following given points-

a) Fatty acid profile should be expressed in SI units i.e. g/kg of total lipid.

b) Units expressed in material and method section should be SI units.

c) Standard notations for units should be used. For instance, author used gr as notation for gram. It should be g.

d) Line no. 128. Cite the reference for DNA extraction method. As the kit used for DNA content analysis is based on someone's protocol/ principle.

e) Line no 135. Is It 1.8% agar or 18% agar. Check it.

f) Line number 169. Expression of fatty acid composition is shown as % total fatty acid content? It should be checked and should be expressed per kg of total lipid.

g) Line number 161- H2, should be changed to H2.

h) Reference should be rechecked and should be in justified alignment.

4. In general, the grammar, punctuation errors should be checked.

5. Authors need to write a profound conclusion.

I am very satisfied with work done by authors and recommending the manuscript after the completion of the revision.

Reviewer #4: This study describes the feeding preferences of a generalist predator in the Antarctic Peninsula after a population collapse following overexploitation. Conclusions surrounding relative abundance of prey species are drawn from quantity of prey found in diet of TOA over two years of fishing. I find this study well written and interesting and would recommend it for publication after minor changes listed below.

Abstract: “Other rare prey found”, infers that the prey listed previously was also rare? I assume that ‘rare’ refers to rare diet choice as opposed to rare species. Please clarify.

Line 54: change to ‘The TOA is by far the…’

Line 67: maybe direct the reader to Fig. 1 here?

Line 79: be clearer with what you mean by ‘short-term feeding’, i.e., opportunistic feeding? That prey was therefore likely just ingested but not assimilated?

Line 80: first time abbreviation SO is used, please define.

Line 83: change ‘its’ to TAO

Lines 87-92: But what is still lacking from these previous studies? Why do this study?

Line 95: change to past tense ‘allowed’

Fig.1: what do the coloured areas (red and yellow hatched lines) represent?

Line 110: change to ‘and therefore were not considered…’

Line 123: what does ‘(mid)’ refer to here? Is this an acronym? If so, please consider capitalising. Same applies to ‘(gid)’ in line 124.

Line 125: was it always possible to identify dorsal muscle from individual prey that have been highly digested?

Line 128: any minimum DNA concentration used before sending?

Line 176-178: consider italicising those that are in the equations

Line 193: change to ‘according to’

Line 210: first time abbreviation is used, please define

Line 213: might be more informative if you report SD without the inclusion of the ∼17% of fish with no stomach contents.

Line 215: where did these results (ρ= 0.29, p < 0.01) come from? I don’t see them in Figure 3. What does ρ refer to?

Line 216: same again, might be more informative if you report SD without the inclusion of fish with no stomach contents.

Line 217: same again, where did these results (ρ= 0.29, p < 0.01) come from? I don’t see them in Figure 3. What does ρ refer to?

Line 218: Specify Fig. 3c,d for 2021 (and 3a,b for 2020 in line 215)

Line 222/Methods section: was it the same person recording how digested prey items were? I.e. was there inter-observer variability in this subjective measurement?

Table 1: missing W% from figure legend, and missing M% from table – mixed up? Would be good to have a thicker line separating the 2 years to make reading the table easier.

Line 272: first time 2019/20 and 2021/22 has been used, try and be consistent

Table 2/Line 244: Although significant, which size class (i.e. G1,G2 or G3) and/or fishing season (2021 or 2021) had the most effect on prey specific numeric abundance? Post-hoc comparisons would be useful here.

Line 244: sex does have an effect but was not included in the best model (as determined from AIC). It is also likely confounded with size class as well, especially in 2021.

Table S2: define what ΔAIC is in the legend.

Fig. 4: would be good to arrange your legend in order of the size classes to make it clearer. Also label which is G1, G2 and G3.

Line 245: were they higher??

Line 314: change to “From these fishes, Macouridae was the…”

Line 323: cephalopoda seem to be a more important prey than Channichthyidae. Maybe change to ‘other important fishes included Channichthyidae’, or something along those lines.

Line 379: change to ‘temperature’

Discussion: would be good to expand upon the specific impacts of climate change on notothenioid species since this is often referred to throughout the manuscript, but only touched upon in the discussion.

Indicate which changes you require for acceptance versus which changes you recommendAddress any conflicts between the reviews so that it's clear which advice the authors should followProvide specific feedback from your evaluation of the manuscriptPlease ensure that your decision is justified on PLOS ONE’s publication criteria and not, for example, on novelty or perceived impact.

We look forward to receiving your revised manuscript.

Kind regards,

Dharmendra Kumar Meena

Academic Editor

PLOS ONE

Journal Requirements:

2. We note that Figure (1) in your submission contain copyrighted images. All PLOS content is published under the Creative Commons Attribution License (CC BY 4.0), which means that the manuscript, images, and Supporting Information files will be freely available online, and any third party is permitted to access, download, copy, distribute, and use these materials in any way, even commercially, with proper attribution. For more information, see our copyright guidelines: http://journals.plos.org/plosone/s/licenses-and-copyright.

1. You may seek permission from the original copyright holder of Figure (1) to publish the content specifically under the CC BY 4.0 license. 

Additional Editor Comments (if provided):

The authors need to improve the article alot in terms of language and comments given by the esteemed reviewers. So I recommends its major revision.

Reviewers' comments:

Reviewer's Responses to Questions

**Comments to the Author**

1. Is the manuscript technically sound, and do the data support the conclusions?

Reviewer #1: Yes

Reviewer #2: Yes

Reviewer #3: Yes

Reviewer #4: Yes

2. Has the statistical analysis been performed appropriately and rigorously? 

Reviewer #1: Yes

Reviewer #2: Yes

Reviewer #3: Yes

Reviewer #4: No

3. Have the authors made all data underlying the findings in their manuscript fully available?

Reviewer #1: Yes

Reviewer #2: No

Reviewer #3: Yes

Reviewer #4: Yes

4. Is the manuscript presented in an intelligible fashion and written in standard English?

Reviewer #1: No

Reviewer #2: Yes

Reviewer #3: Yes

Reviewer #4: Yes

5. Review Comments to the Author

Reviewer #1: have checked the paper entitled " Trophodynamics of the Antarctic toothfish (Dissostichus mawsoni) in the Antarctic Peninsula: Ontogenetic changes in diet composition and prey fatty acid profiles" The paper was very good and well written but needed some corrections as below:

1- For molecular analyses, you reached the phylogenetic tree or only by BLAST, the sequences checked, and what species compared with?

2- Please improve the English language for the paper

3- In Fig. 3. R2 ratio is so weak; do you have any explanation for it?

4- The rest of the comments are implemented in the attached file.

Reviewer #2: Recent study on "Trophodynamics of the Antarctic toothfish (Dissostichus mawsoni) in the Antarctic

Peninsula: Ontogenetic changes in diet composition and prey fatty acid profiles" base on Prey- Predator relationship in aquatic ecosystem is reliable which gives the information about the food chain also. The study of feeding material or prey based on molecular basis is give the trustable impact of the effective study But in graphical representation figure no.3 between in numbers and length is confusable regarding the depending and independing varriable (length,Number) for choosing the cordinates (X and Y). so please check it and see the manuscript which is attach with minor correction in Graph.

Reviewer #3: I have evaluated the manuscript “PONE-D-23-08000 Trophodynamics of the Antarctic toothfish (Dissostichus mawsoni) in the Antarctic Peninsula: Ontogenetic changes in diet composition and prey fatty acid profiles. I am suggesting the authors and curious to know the some major observations-

1. Authors did two sampling in two successive years in the month of February. What was the reason for choosing the month February? Why other months were not chosen for sampling? Whether two samplings are enough to determine the feeding habits of different species? Explain it.

2. Authors dissected out the stomach of different samples and classified the prey items based on the extent of digestion in stomach. Digestion starts in stomach and only protein is digested in stomach. Then how can you have classified them in different extent of digestion? In this scenario, how did you take the sample of these preys? And if prey were gone digestion process, whether it represented the actual fatty composition? Authors have performed fatty acid composition and correlated with nutritional quality of prey. What was the reason behind it? How can authors correlates the fatty acid composition with prey’s nutritional quality.

3. The manuscript is well written and results are well described. Data were well statistically analysed. Results are properly discussed in discussion section. The authors need to revise manuscript in following given points-

a) Fatty acid profile should be expressed in SI units i.e. g/kg of total lipid.

b) Units expressed in material and method section should be SI units.

c) Standard notations for units should be used. For instance, author used gr as notation for gram. It should be g.

d) Line no. 128. Cite the reference for DNA extraction method. As the kit used for DNA content analysis is based on someone's protocol/ principle.

e) Line no 135. Is It 1.8% agar or 18% agar. Check it.

f) Line number 169. Expression of fatty acid composition is shown as % total fatty acid content? It should be checked and should be expressed per kg of total lipid.

g) Line number 161- H2, should be changed to H2.

h) Reference should be rechecked and should be in justified alignment.

4. In general, the grammar, punctuation errors should be checked.

5. Authors need to write a profound conclusion.

I am very satisfied with work done by authors and recommending the manuscript after the completion of the revision.

Reviewer #4: This study describes the feeding preferences of a generalist predator in the Antarctic Peninsula after a population collapse following overexploitation. Conclusions surrounding relative abundance of prey species are drawn from quantity of prey found in diet of TOA over two years of fishing. I find this study well written and interesting and would recommend it for publication after minor changes listed below.

Abstract: “Other rare prey found”, infers that the prey listed previously was also rare? I assume that ‘rare’ refers to rare diet choice as opposed to rare species. Please clarify.

Line 54: change to ‘The TOA is by far the…’

Line 67: maybe direct the reader to Fig. 1 here?

Line 79: be clearer with what you mean by ‘short-term feeding’, i.e., opportunistic feeding? That prey was therefore likely just ingested but not assimilated?

Line 80: first time abbreviation SO is used, please define.

Line 83: change ‘its’ to TAO

Lines 87-92: But what is still lacking from these previous studies? Why do this study?

Line 95: change to past tense ‘allowed’

Fig.1: what do the coloured areas (red and yellow hatched lines) represent?

Line 110: change to ‘and therefore were not considered…’

Line 123: what does ‘(mid)’ refer to here? Is this an acronym? If so, please consider capitalising. Same applies to ‘(gid)’ in line 124.

Line 125: was it always possible to identify dorsal muscle from individual prey that have been highly digested?

Line 128: any minimum DNA concentration used before sending?

Line 176-178: consider italicising those that are in the equations

Line 193: change to ‘according to’

Line 210: first time abbreviation is used, please define

Line 213: might be more informative if you report SD without the inclusion of the ∼17% of fish with no stomach contents.

Line 215: where did these results (ρ= 0.29, p < 0.01) come from? I don’t see them in Figure 3. What does ρ refer to?

Line 216: same again, might be more informative if you report SD without the inclusion of fish with no stomach contents.

Line 217: same again, where did these results (ρ= 0.29, p < 0.01) come from? I don’t see them in Figure 3. What does ρ refer to?

Line 218: Specify Fig. 3c,d for 2021 (and 3a,b for 2020 in line 215)

Line 222/Methods section: was it the same person recording how digested prey items were? I.e. was there inter-observer variability in this subjective measurement?

Table 1: missing W% from figure legend, and missing M% from table – mixed up? Would be good to have a thicker line separating the 2 years to make reading the table easier.

Line 272: first time 2019/20 and 2021/22 has been used, try and be consistent

Table 2/Line 244: Although significant, which size class (i.e. G1,G2 or G3) and/or fishing season (2021 or 2021) had the most effect on prey specific numeric abundance? Post-hoc comparisons would be useful here.

Line 244: sex does have an effect but was not included in the best model (as determined from AIC). It is also likely confounded with size class as well, especially in 2021.

Table S2: define what ΔAIC is in the legend.

Fig. 4: would be good to arrange your legend in order of the size classes to make it clearer. Also label which is G1, G2 and G3.

Line 245: were they higher??

Line 314: change to “From these fishes, Macouridae was the…”

Line 323: cephalopoda seem to be a more important prey than Channichthyidae. Maybe change to ‘other important fishes included Channichthyidae’, or something along those lines.

Line 379: change to ‘temperature’

Discussion: would be good to expand upon the specific impacts of climate change on notothenioid species since this is often referred to throughout the manuscript, but only touched upon in the discussion.

6. PLOS authors have the option to publish the peer review history of their article (what does this mean?). If published, this will include your full peer review and any attached files.

Reviewer #1: No

Reviewer #2: **Yes: **Dr. Veerendra Singh

Reviewer #3: No

Reviewer #4: No

---

## [Author Response · Author response to Decision Letter 0]

25 May 2023

Dear editor and reviewers,

Thank you for your constructive review of our study that definitely contributed to improving the manuscript. Responses below describe changes based on all comments. In the review letter, we refer to the text indicating the lines of the clean version of the MS, rather than the mark-up version. We hope you find our changes satisfactory.

---

## [Editor Report · Decision Letter 1]

6 Jun 2023

Trophodynamics of the Antarctic toothfish (Dissostichus mawsoni) in the Antarctic Peninsula: Ontogenetic changes in diet composition and prey fatty acid profiles

PONE-D-23-08000R1

Dear Dr. Francisco 

We’re pleased to inform you that your manuscript has been judged scientifically suitable for publication and will be formally accepted for publication once it meets all outstanding technical requirements.

Kind regards,

Dharmendra Kumar Meena

Academic Editor

PLOS ONE

Additional Editor Comments (optional):

The article now can be accepted for publication
---

## [Editor Report · Acceptance letter]

8 Jun 2023

PONE-D-23-08000R1 

Trophodynamics of the Antarctic toothfish (*Dissostichus mawsoni*) in the Antarctic Peninsula: Ontogenetic changes in diet composition and prey fatty acid profiles 

Dear Dr. Santa Cruz:

I'm pleased to inform you that your manuscript has been deemed suitable for publication in PLOS ONE. Congratulations! Your manuscript is now with our production department. 

Kind regards, 

on behalf of

Dr. Dharmendra Kumar Meena 

Academic Editor

PLOS ONE